# Membrane Curvature, Trans-Membrane Area Asymmetry, Budding, Fission and Organelle Geometry

**DOI:** 10.3390/ijms21207594

**Published:** 2020-10-14

**Authors:** Alexander A. Mironov, Anna Mironov, Jure Derganc, Galina V. Beznoussenko

**Affiliations:** 1Department of Cell Biology, The FIRC Institute of Molecular Oncology, 20139 Milan, Italy; 2Imaging Facility, Universita Vita-Salute San Raffaele, 20132 Milan, Italy; anna.mironov@yandex.ru; 3Institute of Biophysics, Faculty of Medicine, University of Ljubljana, 1000 Ljubljana, Slovenia; jure.derganc@mf.uni-lj.si

**Keywords:** budding, COP, caveola, endosome, filopodia, membrane fission, Golgi, mitochondria fusion, nuclear envelope, trans–membrane area asymmetry

## Abstract

In biology, the modern scientific fashion is to mostly study proteins. Much less attention is paid to lipids. However, lipids themselves are extremely important for the formation and functioning of cellular membrane organelles. Here, the role of the geometry of the lipid bilayer in regulation of organelle shape is analyzed. It is proposed that during rapid shape transition, the number of lipid heads and their size (i.e., due to the change in lipid head charge) inside lipid leaflets modulates the geometrical properties of organelles, in particular their membrane curvature. Insertion of proteins into a lipid bilayer and the shape of protein trans-membrane domains also affect the trans-membrane asymmetry between surface areas of luminal and cytosol leaflets of the membrane. In the cases where lipid molecules with a specific shape are not predominant, the shape of lipids (cylindrical, conical, or wedge-like) is less important for the regulation of membrane curvature, due to the flexibility of their acyl chains and their high ability to diffuse.

## 1. Introduction

Biological membranes consist of two lipid monolayers, attached to each other through acyl chains. Each leaflet functions as an independent two-dimensional fluid. The liquid mosaic hypothesis about structure and functioning of trans-membrane proteins in biological membranes poses that trans-membrane proteins float in a liquid lipid bilayer [1,2,3]. The lipid bilayer is not compressible when the compression force acts along the membrane plane, but bends easily. When the membrane is stretched, a hole is easily created in the membrane [2,3,4]. At a certain point of the membrane surface, the degree of membrane bending is described in terms of the membrane curvature, and determined by local asymmetries in the size of the areas of two bilayer leaflets [3,4,5,6]. Here, for simplicity of consideration, we use only ideal models, and deal with the spontaneous membrane curvature, which depends on the bilayer itself.

On the other hand, it is necessary to take into consideration how and when measurement of the membrane curvature is performed. For instance, when one measures the level of membrane curvature by means of electron microscopy, the results depend on the sample preparation, namely, fixation and staining procedures. For instance, contrasting with osmium makes projection of the membrane visible in the image thicker. After cell treatment with OsO4, the thickness of the membrane projection is equal to 9 or even 10 nm. As a result, the difference between the areas of the outer layer and the inner layer will be higher. For instance, in living cells the trans-membrane area asymmetry (TAA; the ratio between the surface area of two membrane leaflets) of the coatomer (COP) I-dependent vesicles is 1.4, whereas after fixation by osmium OsO4 the measurement of this parameter would give 1.5. In living cells, the TAA of synaptic vesicles with a diameter of 44 nm is equal to 1.5. After fixation with OsO4 and embedding it increased up to 2.5. On the other hand, the mean diameter of intra-luminal vesicles inside multi-vesicular bodies is 42 nm. Their TAA is equal to 1.53; after the preparation of samples for electron microscopy TAA will be equal to 2.61 [4] (Figure 1A,B).

## 2. Factors Determining the Membrane Curvature

The shape of the lipids is important for the stability of this lipid bilayer and membrane curvature. The speed of the generation of conical or wedge-like lipids by proteins interacting with lipid molecules (BARS and others) is a slow process, and is limited by delivery by diffusion of the lipids suitable for transformation. The shape of the lipids (cylindrical, conic, or wedge-like) is also less important because carbon chains can bend (Figure 1C,D). Bending the lipid bilayer forces chains of fatty acids within the leaflet, present on the convex side, to straighten, and turn the chains into a straight line. On the contrary, on the concave side, they should be maximally bent to fill the internal membrane volume. However, the role of the addition of new lipid molecules into the bilayer, or the change of lipid shape due to their enzyme-dependent modification, is rather limited because of the rapid diffusion of lipids and their intermixing. Thus, shape of lipid has a low importance for the generation of local membrane curvature. In order to amplify local curvature, protein insertion into the lipid bilayer should be synchronous and fast [3]. Moreover, accumulation of cone-shaped lipids in one place would hinder the diffusion of cylindrical lipids to the site of the transformation, even if the rate of their formation was high [7]. A cluster of the conical and wedge-shaped lipids cannot change the shape of the organelle because the lipids tend to diffuse. Formation of tubules by large membrane vesicles is observed when the vesicles are forced to transfer part of their lipids from the inner to the outer surface [8].

However, the actual membrane curvature could also be imposed by external forces not related to lipids. Flip-flop of lipid molecules also affects membrane curvature [9,10,11]. In artificial membranes, the spontaneous flip-flop of lipids with large or charged head groups, glycolipids, and phospholipids, is very slow (t_1/2_ of many hours or days), whereas molecules like fatty acids, cholesterol, diacyl glycerol, and ceramide flip in seconds [12,13]. In biological membranes the lipid translocation is accelerated to a biologically relevant rate by protein catalysts, denoted by the general term flippases [14,15]. When the capacity of flippases to move lipids from one leaflet of lipid bilayer to another is limited, the transformation of the organelle shape depends on the low TAA. Often, they are called flippases if they move the lipids to the cytoplasmic side, floppases if they move the lipids in the opposite direction, and scramblases if they transport the lipids in both directions. The activity of the transporters may, or may not, depend on ATP. For example, in the ER membranes extracted from rat liver a fast (t_1/2_ on the order of several seconds), ATP-independent redistribution of phosphatidylcholine and phosphatidylethanolamine between the membrane leaflets could be observed [16,17]. Flippases and even simple flip-flops are important for correction of TAA [18,19]. In addition, lipid molecules can be added to the cytosolic leaflet through aqueous phase with the help of specific (i.e., phospholipid transfer protein) or nonspecific (i.e., albumin and non-specific lipid transfer proteins) lipid transporters [20,21,22,23].

Although the membrane curvature is important, one should understand, that the force generated by membrane tension is rather week. In contrast, forces generated due to the changes in the volume and surface area are typically stronger than the forces generated according to the TAA. For instance, forces generated by changes in osmotic pressure corresponding to water movement are significantly higher than the forces responsible for the prevention of the contact between hydrophobic acyl chains and water (this is a real molecular mechanism, on which the TAA hypothesis is based). If the osmotic pressure on both sides of the membrane is equal, then difference between these forces is equal to zero. The local membrane curvature affects the organelle shape when volume and surface area are constant. However, when water moves to the lumen, under the action of osmotic forces, the membrane structure cannot preserve its initial shape (the vacuole is transformed into a sphere) and then even its integrity. For instance, when artificial vacuoles are made in hypertonic solution, and then placed into hypotonic solution, water molecules begin to move into the lumen, causing transient lytic membrane holes, and fluid starts to go out of the vacuole [24]. The enrichment of lipids, which is able to induce membrane curvature in the rims of Golgi cisternae, cannot counteract other membrane forces [2]. In theory, undulation of the membrane could compensate for alterations of the membrane curvature. However, these undulations need energy consumption, because for this it is necessary to move cytosol (Figure 1E–J). Therefore, in quickly frozen cells the shape of the organelles tends to be almost ideal [25].

An important aspect of membrane geometry in cells is the speed of the curvature delivery/extraction during fusion of membrane organelles, and the speed of alterations in their volume and surface area. For instance, the pulling tether is stable when it is formed slowly, and can be disrupted at higher speed [26,27,28,29]. Experiments with quick freezing of synapses, after their stimulation, demonstrated that lipid physics is responsible for these extremely fast formations of buds, on the presynaptic membrane after the fusion of synaptic vesicles with it [30,31]. In a few milliseconds after the fusion of a few synaptic vesicles with the pre-synaptic membrane, formation of domed (hemisphere-like) invagination (hemispheres and buds), with a diameter of 80 nm was observed. Integration of the synaptic vesicle into the pre-synaptic membrane leads to a very rapid formation of invagination, with the shape of hemispheres. This phenomenon occurred much faster than could be done by diffusion of proteins, which form a membrane coat. Integration of a synaptic vesicle into a pre-synaptic membrane leads to in excess of 1620 molecules of lipids (calculations are performed on the basis of data by Takamori et al. [32]) within the cytosolic leaflet of the pre-synaptic membrane. The trans-membrane area difference of one hemisphere, with a two-fold higher diameter (80 nm) than the diameter of a synaptic vesicle (42 nm), equals the one of a synaptic vesicle. These calculations support the trans-membrane area asymmetry and curvature redistribution hypothesis (TAA hypothesis).

External forces may significantly change the equilibrium membrane shape. For instance, proteins may also affect the local membrane properties as external factors. Membrane-bound proteins such as coatomer (COP) I, COPII, dynamin, clathrin/AP1, clathrin/AP2, AP3, AP4 (reviewed in [26,33,34,35]), with asymmetric distribution of mass across the bilayer, influence membrane bending in a nonspecific manner due to molecular crowding [36]. Shallow hydrophobic insertions and strongly curved protein scaffolds appear to be effective generators of the membrane curvature of intracellular organelles [37]. Curved proteins may bind to the membrane and also impose their curvature onto the membrane [5]. It has been suggested that the binding of crescent-shaped BAR domains to the membrane plays a role in the formation of membrane tubules, and that the resulting spontaneous membrane curvature might depend on the fraction of the membrane surface area covered by BAR [36]. However, insertion of one protein helix would affect the curvature only slightly. In order to increase local curvature significantly, it is necessary to insert several such proteins in one place, and to keep these proteins close to each other, otherwise they will be dissolved within the membrane fluid. Thus, these proteins should be concentrated, by some mechanism, within one place [36]. The curvature of BAR-like domains is often lower [2] than that of most of membrane tubules in the cytosol (Figure 2A). If the curvature of BAR-like domains is lower than that of a membrane tube, the proteins orient from the perpendicular position to a more longitudinal one. The curvature of the projection of its inner surface onto the long axis of the tube (when viewed along the long axis) increases. Finally, the tube becomes so thin that even when this protein is positioned at a very small angle from the long axis of the tube, the curvature of its inner surface to the long axis of the tube becomes too large for this curvature, and the protein detaches from the tube inducing tubule fission (see below) (Figure 2B–G).

When insertion of proteins into the lipid bilayer occurs slowly, formation of membrane buds can be rather rare, because entropy-dependent diffusion of these proteins is faster than their insertion. Local augmentation of membrane curvature can occur due to sudden insertion or subsequent elimination of protein helices into the cytosolic leaflet [26]. The shape of the trans-membrane domains of proteins also affects the membrane curvature. Differences between the length of protein trans-membrane domains and the bilayer thickness lead to the so-called hydrophobic mismatch, which leads to immiscibility within a single phase or interaction of lipid with integral or peripheral proteins, and induces lateral segregation of membrane proteins [37]. The separation of membrane lipids into compositional domains can be due to lateral phase separation. For instance, when a transferrin receptor was incorporated into an artificial lipid bilayer, the conformational changes of the latter were observed, only if the lipid bilayer was composed of a mixture of different lipids, but did not occur if the lipid bilayer was composed of homogenous lipid molecules [38,39]. The inter-compartmental barriers are sufficiently low and narrow [40]. Furthermore, glycocalyx regulates the membrane curvature, especially in endothelial cells [41].

We tested the role of the TAA hypothesis in the shaping of cell organelles using three different experimental models, namely: The fusion of small spheres with discs with toroid edges, or the separation of a small sphere from such discs; the fusion of two spheres; and finally, the separation of the cylinder into spheres, when the diffusion of lipids into the cylinder is limited due to low temperatures [4]. These experiments have a common feature, namely, the delivery of lipid molecules by their rapid diffusion along one of the leaflets biological membranes is limited or severely limited. In all these cases, the predictions obtained from mathematical models were fulfilled. Extraction of membrane curvature from Golgi membranes by COPI vesicles affects the shape of this organelle. Similarly, extraction of highly curved domains from membranes of multi-vesicular bodies, by intra-luminal vesicles, affects the shape of the organelle. In turn, the Golgi tubulation is critically dependent on the delivery of curvature by COPI-dependent vesicles, after their fusion with Golgi cisternae, and vice versa, the extraction of membrane curvature into 52 nm vesicles, is accompanied by the transformation of Golgi tubules into cisternae.

## 3. The Golgi Complex and the Role of TAA in the Sorting Functions of this Organelle

The Golgi complex (GC) is one of the most complicated membrane organelles. The membrane curvature redistribution hypothesis predicts that budding and consecutive fission of COPI-dependent Golgi vesicles with a high level of their curvature, transforms a highly perforated Golgi cisterna with undulated rim into a cisterna, with narrower openings and less undulated rims. When the level of undulation becomes minimal, further detachment of COPI vesicles transforms a solid disk into a solid invaginated semi-sphere. In contrast, the fusion of COPI-dependent vesicles with a Golgi cisterna would widen cisternal pores and increase the level of rim undulation [4].

Additionally, we demonstrated that (i) inhibition of the SNARE machinery alone reduced TAA of the Golgi cisternae, and induced a narrowing of the cisternal perforations, followed by invagination of cisternal membranes; (ii) inhibition of the ARF/COPI machinery alone increased the TAA of the Golgi cisternae, and induced a widening of the cisternal perforations, followed by the Golgi tubulation (it was known that inhibition of the activity of the ARF/COPI machine led to the Golgi tubulation when GC is functionally isolated from the ER, whereas otherwise Golgi markers go to the ER [4]); and (iii) inhibition of both machineries did not change Golgi shape significantly. In all of these cases TAA remained constant (iv). Diminution of the volume of the Golgi cisternae did not induce a significant change in TAA (v). Estimation of TAA also tells us that it is impossible to obtain stacked Golgi by fusion of small vesicles. In experiments by Rabouille et al. [42], it was not possible to obtain the formation of Golgi stacks from vesicular fraction, generated during Golgi fragmentation by mitotic cytosol. This reaction needs the addition of tubular/cisternal fraction and vesicles. Of interest, Golgi vesiculation quickly goes to a plateau [33], indicating that the available TAA is finished.

When the surface area and the volume of the Golgi cisterna are constant, its TAA can be increased by undulation of cisternal rims (Figure 1E–J). Moreover, our calculations suggest that when the volume of a thick disk with toroidal rims is decreased two-fold, TAA would be almost unchanged. The maximal change of TAA, in the case of a disk with diameter of 1 µm and thickness 40 nm, would be 0.01%. High curvature of Golgi membranes facilitates their deformation. The force needed to extend tethers from ER membrane networks formed in vitro is 18.6 ± 2.8 pN, whereas the force necessary to make a tether from Golgi membrane networks formed in vitro is equal to 11.4 ± 1.4 pN [43]. One of the biological consequences of this feature could be the following: COPI-dependent vesicles would be more easily formed on Golgi membranes than on membranes of the ER. Indeed, blockage of membrane fusion mostly induces vesiculation of the GC, whereas vesiculation of ER exit sites is modest [44], although COPI is present there [45].

The TAA hypothesis predicts that the GC containing a smaller number of vesicles should be more resistant to Brefeldin A (BFA) [5,46]. For instance, when we incubated isolated a Golgi membrane containing the cisternae and almost no vesicles the Golgi tubulation was not observed. In contrast, after the addition of vesicles the GC became tubulated [4]. Similarly, in cells devoid of microtubules, the number of 52-nm vesicles around the Golgi mini-stacks is significantly smaller than the number around the central GC in control cells [46]. As a result, the redistribution of Golgi enzymes into the ER is slower than in control cells [47]. Moreover, due to absence of small vesicles, isolated Golgi membranes are more resistant to the action of BFA than Golgi membranes in the presence of such vesicles [4,48]. A discrepancy between the results of Misteli and Warren [49], and Orci et al. [48], who described the tubulation of isolated Golgi membranes after inhibition of the ARF/COPI machinery, and of Happe et al. [50], who did not observe Golgi tubulation, is caused by the following reasons: In experiments by Happe et al. [50], the Golgi cisternae were initially attached to mica and incubated with cytosol. Under these conditions, COPI 52-nm vesicles are not present in the solution and cannot deliver curvature to the Golgi cisternae [4]. In contrast Misteli and Warren [49,51] used the full Golgi preparation, which contained COPI vesicles.

The GC is the organelle where coat proteins play one of the most important roles. When insertion of proteins into the lipid bilayer occurs slowly, formation of membrane buds can be difficult because, due to entropy, diffusion of these proteins away, will be faster than their insertion of the second, third, and so on protein molecules. Coat proteins interact with membranes partially inserted into the cytosolic leaflet. The dissipation of coat proteins is prevented by their polymerization. With time this can induce vesicle fissions. Yang et al. [52] demonstrated that COPI vesicle fission occurs according to two sub-stages: (1) an earlier stage of bud-neck constriction, in which BARS and other COPI components are required, and (2) a later stage of bud-neck scission, in which phosphatidic acid generated by phospholipase D2 is also required. It seems that COPI per se cannot induce fission, an additional mechanism is necessary. For instance, COPI- coated buds generated in the presence of GTPgS are resistant to uncoating [53]. Additionally, in the absence of functional COPI [33], (when COPI suddenly loses its ability to form COPI-coated buds and vesicles), detachment of COPI from the membrane becomes less efficient [54,55]. Furthermore, if it is not possible to form membrane curvature, and then induce COPI uncoating due to fission of COPI-dependent vesicles the turnover of COPI decreases [33/25]. On the basis of these findings Zhukovsky et al. [26] proposed that proteins containing amphipathic helices mediate membrane fission via shallow insertion of these helices into the lipid bilayer. Such proteins require one or two specific lipid cofactors to generate membrane curvature and to complete the fission process [56].

Generally, the concept of protein concentration as a way to modify the membrane curvature, especially in the frame of the GC, the main sorting station platform in the cell and the most important site of protein glycosylation, is key for the understanding of the function of the GC and its reorganization in pathology [57]. Coat proteins and matrix protein, together with Golgi membrane proteins, determine the Golgi morphology. The crowded protein environment on the surface of cellular membranes can contribute to membrane shape change [58]. This issue deserves additional analysis.

In contrast, we propose that when several helices, inserted into the lipid bilayer, are situated around the thin membrane tubules, for example within the neck of the membrane bud, their sudden elimination from the membrane would induce the formation of an excess of lipid head within the luminal leaflet, due to the fast filling of the empty space by heads of surrounding lipid molecules (Figure 3A–C). This excess makes the channel thinner and stimulates periplasmic fusion, and the tubule undergoes fission (Figure 3D–G). This excess could be dissolved easily if the membrane curvature was not so high. In contrast the diffusion of lipids along the thin tubule is rather slow, and this could induce destabilization of the lipoid bilayer and fission of the bud with the formation of a vesicle [6]. However, this hypothesis needs to be supported by some data or modeling.

On the other hand, formation of protein coats responsible for the formation of small vesicles could be formed according to the following model (Figure 2H–O). Initially, a small part of one protein molecule is introduced into the cytosol monolayer of the flat membrane. This results in the area of the cytosol monolayer being larger than that of the luminous one. As a result, two paths are possible. 1. The flat membrane bends locally towards the cytosol, forming the initial stage of the membrane bud. 2. Part of the lipid molecule making up the cytosol monolayer is shifted from this place, and redistributed along the flat membrane making it slightly convex towards cytosol. In the first case, when the second molecule is introduced into the membrane in the same way, a more protruding kidney emerges; if these two molecules do not bind to each other by themselves, or by means of the second protein or protein complex, the system passes to the second path because of entropy.

If the molecules of the first protein bind with each other by the second molecule, for example, by COPI or a triscoledon of clathrin coat, the convexity is fixed on the membrane. After the fixation to each other of two molecules of the first protein by the second protein, the third molecule of the first can penetrate into the monolayer, especially since on the convex surface of the membrane lipid heads are less tightly pressed against each other. Then the third molecule of the first protein fixes the molecule of the second protein again, making the membrane kidney protrusion. In this case, the molecule of the first protein, being in the center of the coat, is not necessary and can be detached. So step by step the membrane kidney is formed, the curvature of which is determined by properties of the third protein, namely, how rigid it is. Finally, the stable rigid coat, which is attached to the lipid bilayer only at its edges, is formed (Figure 2O). In mature buds, the coat is attached to the neck of the bud (Figure 3D).

The TAA hypothesis explains why the number of ARF1 in one COPI-coated bud is significantly lower [59,60] than it should be, according to the model describing formation of COPI coat [61]. Renard et al. [62] classified membrane fission mechanisms into two main categories: active fission (with the direct consumption of cellular energy by nucleoside triphosphate hydrolysis) and passive fission (without the direct use of energy). Sudden elimination of protein from the lipid bilayer could be due to time-dependent tubule thinning or due to the hydrolysis of GTP, leading to the quick change of the protein conformation.

Another example of in vivo situation where the membrane curvature hypothesis works well is formation of intra-luminal vesicles [3,4]. This hypothesis could explain the contradictions between our results, demonstrating that the diameter of a clathrin-dependent vesicle in yeast is 40 nm [35], and data by Kukulski et al. [63] who observed that the diameter of the invagination of the plasma membrane (PM) with clathrin cap is only 32 nm. According to our membrane curvature hypothesis, if the spherical endosome is constricted by an actin-myosin ring, there should be a break of the sphere during its transition into two spheres. Squeezing fluid from endosomes is seen in video 5 by Liebl and Griffiths [64].

The above-mentioned regularities could be useful for understanding the mechanisms responsible for membrane tube transformation into vesicles through the bead-like tube stage (Figure 4A–D). According to the TAA hypothesis, when the delivery of surface area and curved membrane domains is absent the diameter of formed vesicles will be 2.2-fold larger than the diameter of the original tube. In this way, the breakdown of the cell microvilli, with diameter of 100 nm, into spherical vesicles produced vesicles with a diameter of 198 ± 7 nm [4]. If we take this approach and calculate the diameter of coatomer II (COPII)-dependent vesicles, using data from Bacia et al. [65] as a starting point for this calculation, the diameter of COPII vesicles will be 85 nm. The diameter of vesicles visible within the ER exit sites, and separated from other membranes is 50 nm, whereas the cross diameter of COPII coated buds on the granular ER is about 75 nm [45,66]. Even in the cornerstone paper by Barlowe et al. [67], vesicles do not form after incubation of microsomes with a purified component of COPII. The vesicles appear only after harsh filtration through a gel with narrow pores. This suggests that these 50-nm vesicles are not COPII-dependent.

## 4. Membrane Geometry and Organelle Dynamics

Membrane geometry is important for the understanding of several phenomena in cell biology. One of them is related to the formation of filopodia (Figure 1K,L and Figure 4E,F). If a bundle of actin microfilaments is formed, it begins to grow into an extracellular medium. When filopodia form a cylindrical protrusion from the PM, its geometry is energetically unfavorable [2,3]. A point force would eventually extrude a tether, although not immediately [68]. The reason is the following: When the bundle of actin filaments starts to grow this bundle pulls the PM, a cone should form. However instead of a cone, a thin cylinder and ring below it are formed. For instance, let us consider that the diameter of the filopodia is 100 nm and its length is 2000 nm, and the ring of basis has a radius of 2000 nm. In this case the sum of the surface area of the cylinder and the basic ring would be 1.4-fold smaller than the lateral surface area of the cone, which has the height of 2000 nm and a radius of its basis equal to 2000 nm (Figure 1K,L). Thus, in order to attach PM to the actin bundle there is the necessity of ezrin. Ezrin attachment to the bundle and then to the membrane overcomes this energy problem, and filopodia (cylinder) is formed. For instance, ezrin is enriched in microvilli [69]. If the filopodia meets an obstacle, the membrane grows inward to form inversion (our unpublished observations).

On the other hand, taking into consideration ideal geometrical figures, one could also calculate that after cell division the surface area of the two daughter cells is 2.27-fold higher than the surface areas of the original cell before its division. This membrane should be kept somewhere. Indeed, during mitosis the cells became more spherical, with many processes of the PM, and the nucleus acquires spherical or ovoid shape [70]. Moreover, there should be enough membrane for the series of consecutive divisions, such as formation of blastomers [71]. This suggests that during the G2-phase of the cell cycle, cells should synthesize a significant amount of lipids. Animal cell cytokinesis requires vesicle delivery with their negative (in comparison to the membrane furrows dividing cells) curvature to the site of abscission [71]. Furthermore, if the delivery of new membrane (and its surface area) and synthesis of lamin are slower than the process of the formation of new surface area during cell spreading, the nuclear envelope (NE) would be broken [72]. Indeed, NE consists of a endoplasmic reticulum (ER) membrane and the layer of lamin at the nucleoplasmic surface of the internal membrane of NE. NE is connected with the ER and thus, delivery of additional membrane occurs through these connections, which are rather narrow.

Membrane geometry is important for understanding of the mitochondria transformation in live cells. Mitochondria can fuse and then undergo fission depending on cell cycle and environmental conditions [73,74,75,76,77]. In these reviews on the fusion and fission of mitochondria, the question of the inner and outer mitochondria membranes’ (IMM and OMM) geometry is usually not addressed. Fusion of mitochondria was described in live cells, for instance, during the cell cycle and after starvation and during cytokinesis [78]. After their fusion, the excess of OMM is generated (Figure 5). After the fusion of two mitochondria with a 500-nm diameter and a 1-µm length of its cylindrical part, about 0.1 µm^2^ of excessive membrane would appear. Under normal conditions so-called mitochondria derived vesicles are formed from this additional membrane of OMM. These vesicles go to lysosomes. At high magnification these vesicles appear as multilamellar organelles [78,79]. TOMM20-positive vesicles (this is the indication that these vesicles are derived from OMM, although the labeling density of these proteins is rather low) are found in normal cells after its total 3D reconstruction [80].

On the other hand, after mitochondrion fusion, a septum is formed, which is connected to IMM along the entire mitochondrion circle. This septum prevents diffusion of ions and other substances through the matrix of mitochondria. This is probably why a vast majority of cristae in mitochondria do not have this structure. It is not known how the septum is perforated and which proteins are involved in this process. In addition, the question of restructuring the geometry of IMM and OMM during the fission of mitochondria has not been close to completely addressed. Cristae can have the form of a fold of IMM with a very narrow space (its thickness is 10–15 nm) between two membranes. This fold protrudes into the matrix. On the other hand, cristae can have the form of a plate in which two membrane derivatives of IMM are closely (space thickness is 10–15 nm) attached to each other. These cristae are usually connected with IMM at two or more places, and do not interfere with the diffusion of substances along the mitochondrial matrix. It is known that fission of IMM occurs earlier than the separation of OMM [77]. We believe that this septum could be the place where the IMM and volume of the mitochondrial matrix are separated into two parts. Therefore, if the septum has already been perforated, the closure of these pores by periplasmic fusion is required. If the new septum were formed on the basis of the crista of the second type, it is geometrically, and probably energetically, more advantageous than when the new septa is formed on the basis of the crista of the first type, because less energy-consuming movement of the dense matrix is required.

Another issue important for the mitochondrial fission is the delivery of lipids for the formation of additional surface area of the OMM and IMM. Although lipids are synthesized in the smooth ER, free fatty acids cannot travel through cytosol if the membranes are separated [81,82,83]. The ER-mitochondria contact-sites could be important for the delivery of the membrane necessary for mitochondria fission. Transport of free fatty acids through ER-mitochondria contacts is shown by Rambold et al. [81,84]. For instance, there is no background of the Red-C12 fatty acid in Figure 1B by Rambold et al. [84]. Moreover, pulling one cholesterol molecule (or free fatty acid) out of a lipid bilayer into water required an energy barrier of 80–90 kJ/mol, corresponding to hydrolysis of about 1.5 ATP molecules. Thus, only lipid transfer proteins can deliver cholesterol and free fatty acids from one membrane to another. Numerous (now more than 131) lipid- and sterol-transfer proteins have been identified to mediate directional cholesterol transfer at membrane contact sites formed between two closely apposed organelles (reviewed by Mironov et al. [85]). However, it not known how lipids are moved from OMM to IMM.

Finally, an interesting issue for the TAA hypothesis is the role of caveolae. Smooth muscle cells, cardiomyocytes, and striated myotubes could be in two states: (1) relaxation and (2) contraction. In all these cases the cell surface area and volume are more or less constant, whereas the shape of cells is changed, and after contraction the ratio between the surface area and the volume becomes higher. Thus there should be a mechanism for the reorganization of the cell surface in order to place excessive surface area in specific structures (Figure 6). Theoretically there are two possibilities: (1) simultaneous formation of filopodia and invaginations, or (2) extraction of the excess of the lipid bilayer in the form of caveolae. The first mode needs significant movement of cytosol and reorganization of the membrane curvature pattern. The second mode is more energetically favourable. Although cytosol is rather viscous, interstitium is even more rigid for the filopodia formation. When arteries were fixed with perfusion fixation under normal pressure, the caveola density in endothelial cells and in smooth muscle cells was lower than after the fixation by immersion (Figure 6; see also [86]). Upon stretching of the plasma membrane, caveolae become flat and liberate cavins into the cytosol, and caveolins into the surrounding membrane [87,88,89]. Upon hypotonic-medium induced swelling of skeletal muscle fibers, caveolae disappear [89]. In contrast, after treatment of cells with hypertonic medium, caveolar clustering increases [90]. Of interest, are the shapes of caveolae and caveolar derivates used by Golani et al. [91] observed in tissues only after chemical fixation. In contrast after quick freezing, caveolae do not exhibit necks [92,93].

Caveolae protect the plasma membrane from rupture in the course of cell stretching by external forces such as, not only osmotic pressure, but also by cell–cell and cell–substrate interactions during contraction and relaxation [88]. For instance, if one takes a cube and transforms it into a parallelepiped with half the height of the cube but with the same volume, the surface of the parallelepiped will increase by 17%. If the height is also reduced by 4-fold with the preservation of the same volume, the surface of a thin parallelepiped will be 75% larger than that of the cube. A lot of caveolae were found in muscle cells in the type I pneumocytes of lung, and in some endothelial cells, namely, in the endothelial cells of the arteries, and blood capillaries of the lungs, muscles, and adipose tissue. Glancing sections across the surface of an endothelial cell can show caveolar superstructures, termed clusters or rosettes. It is estimated that the presence of caveolae increases the surface area of the PM of endothelial cells (ECs) by up to 70% [92]. In contrast, in the EC of brain vessels, including blood capillaries, where arterial pulsations are smoothed out, the number of caveolae is rather small in comparison to those in the muscles and lungs (Figure 6). This suggests the role of caveolae as the mechanism for the reservation of the PM surface area. This process does not contradict the curvature hypothesis because insertion of caveolin and cavins into the lipid leaflet from the cytosol side increases the TAA and stimulates the PM invaginations into the cytosol. However, in an adipose tissue, caveolae and their derivate could be mostly important for temporal storage of fatty acids and cholesterol [85]. Thus, formation and disassembly of caveolae could be a dynamic process, and they consume the excess of the PM surface area after contraction of cells.

## 5. Conclusions and Perspectives

The Golgi complex is a central transport and sorting station. Its membranes are very complicated and have highly curved domains. Membrane curvature plays a significant role in phase transition facilitating fission of membrane tubules and making lipid and protein sorting easier. Knowledge about the regularities of membrane geometry and lipid physics are also important for understanding of the mechanisms of synaptic vesicle fusion, with a pre-synaptic membrane and consecutive kiss-and-run events; shaping of membrane organelles such as the GC, endosomes and multi-vesicular bodies; explaining why intraluminal vesicles could be formed in the absence of function of the ESCRT machine [94]; and the brush border in specialized epithelial cells, for instance, of microvilli of kidney epithelial cells of proximal tubules during their preservation in the solution of intracellular type on ice. The membrane curvature-redistribution hypothesis provides new insight into the formation of protein-coated buds and their consecutive fission, with formation of vesicles. Similar approaches suggest that a quick change of nuclear shape could induce breaks of the nuclear envelope after sudden change of the shape of the nuclear envelope, i.e., during movements of cells through narrow channels. Similarly, the fusion of mitochondria generates an excess of the surface area of the outer mitochondrial membrane, whereas their fission needs the delivery of new lipid molecules to the outer mitochondria membrane through ER-mitochondria contact sites [95,96,97].

## Figures and Tables

**Figure 1 ijms-21-07594-f001:**
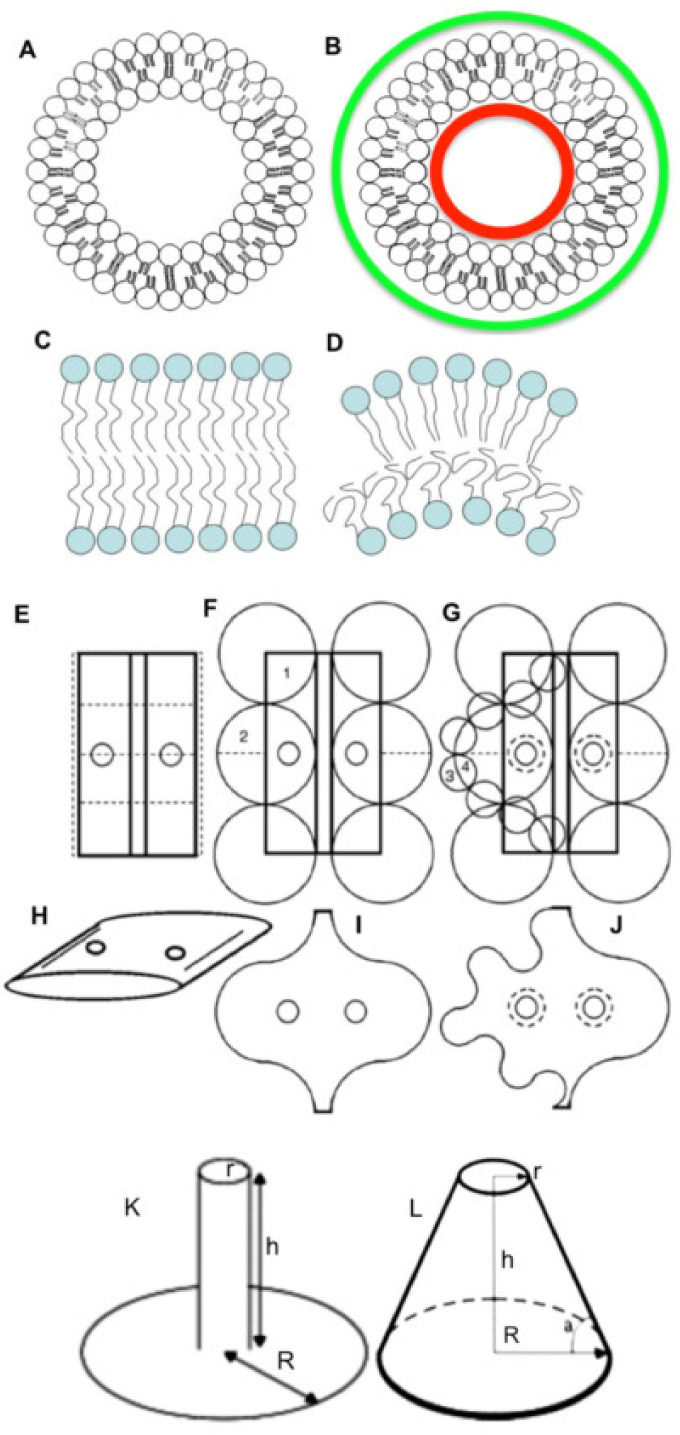
Dependency of membrane curvature on different factors. (**A**,**B**) The role of sample preparation for the results of the measurement of membrane curvature. When the surface area of the leaflets is measured along the lipid head, the trans-membrane area asymmetry (TAA) of the vesicle will be 36/24 = 1.5, whereas if the surface area is measured along the external/internal border of the image visible after treatment with OsO4 (shown by green and red lines) TAA green/red will be 2.5. (**C**,**D**) Membrane bending affects the shape of acyl chains, but the lipid heads are stable. During bending the excess of lipid heads appears within the convex leaflet. (**E**–**J**) Undulation of the Golgi cisterna rims increases the cisterna TAA without significant change of its volume and surface area. Undulation could be large (**F**,**I**) and small (**G**,**J**). (**K**,**L**) Difference in the surface area between the filopodium and truncated cone. Filopodium (**K**): The length equals 2000 nm. The diameter of the filopodium equals 100 nm. The diameter of the plasmalemma surface area, used for the formation of the filopodium, is equal to 4000 nm. The surface area is equal to 1.82 × 10^7^ nm^2^. Truncated cone (**L**): The diameter of the tip is equal to 100 nm. The height is equal to 2000 nm. The diameter of the plasmalemma surface area used for the filopodium formation is equal to 4000. The surface area is equal to 1.32 × 10^7^ nm^2^ (R—radius of the plasmalemma area; r—radius of the tip and filopodium; h—height).

**Figure 2 ijms-21-07594-f002:**
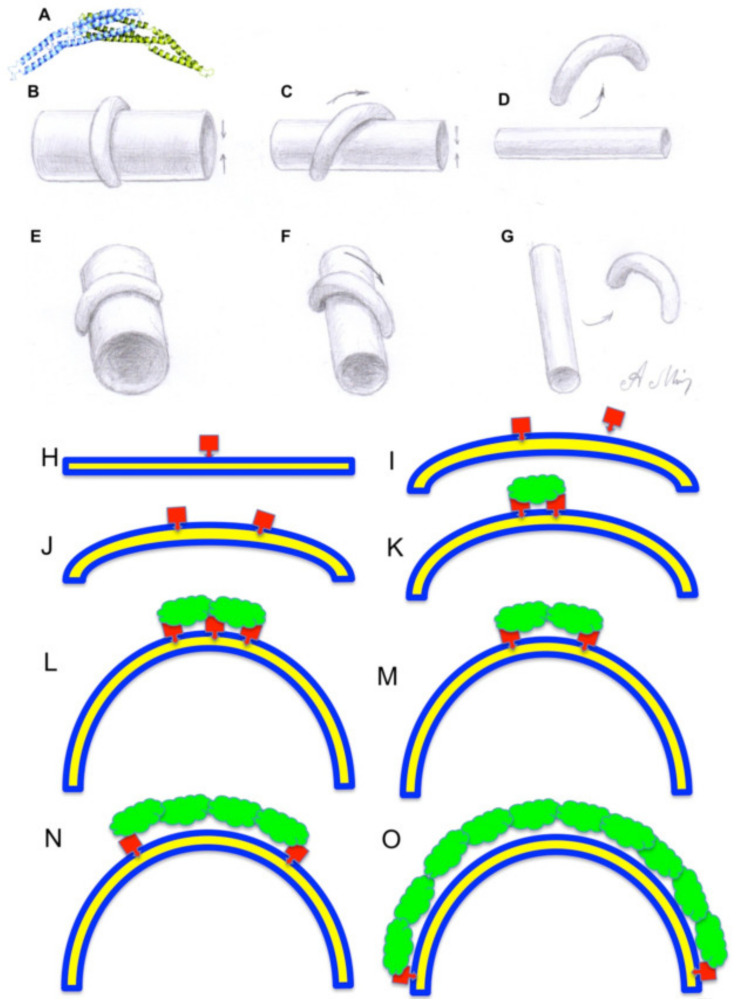
Role of the protein–membrane interactions in membrane curvature. (**A**) An example shape of a BAR domain. (**B**–**G**) Effect of BAR-like domains on membrane tubules and their interaction. Proteins containing BAR-like domains and other similar proteins are not highly bent. The TAA membrane curvature hypothesis suggests the following scenario. After attachment of such proteins to a rather thick membrane tubule, and insertion of part of the protein into the cytosolic leaflet of an endomembrane, proteins mostly orient perpendicularly to the long axis of the tube. Then the tubule becomes thinner, and this BAR-like domain reorients its position from a perpendicular to a more longitudinal one (**F**). Finally, the tubule becomes very thin and its curvature appears higher than the maximal curvature of the domain projection, even at a very small angle to the long axis of the tubule, (**G**) even when the orientation of this protein is at a very small angle related to the long axis of the tubules. As a result, this protein detaches from the membrane. This could lead to the excess of the luminal leaflet inducing the periplasmic fusion and fission of the tube at this site (see Figure 3). (**H**–**O**) A possible model of membrane budding induced by the membrane coat. (**H**) Insertion of a part of the one protein (red) into the lipid bilayer (blue) induces initial membrane bending. (**I**,**J**) Insertion of the second molecule of the first protein into the membrane increases its bending. (**K**) Connection of two first molecules by the second coat protein (green) makes this bending stable. (**L**) Insertion of the third molecule of the red protein, and then its attachment to the already formed coat, increases membrane bending. (**M**) After formation of a rather stable coat the middle molecule one of red proteins can detach. (**M**,**N**) Consecutive augmentation of the surface area of the rigid coat increases the curvature, and makes the coated membrane bud. This bud is attached to the membrane only near the edge of the coat (**N**). This coat is unstable and sensitive to the detachment of red proteins (see Figure 3).

**Figure 3 ijms-21-07594-f003:**
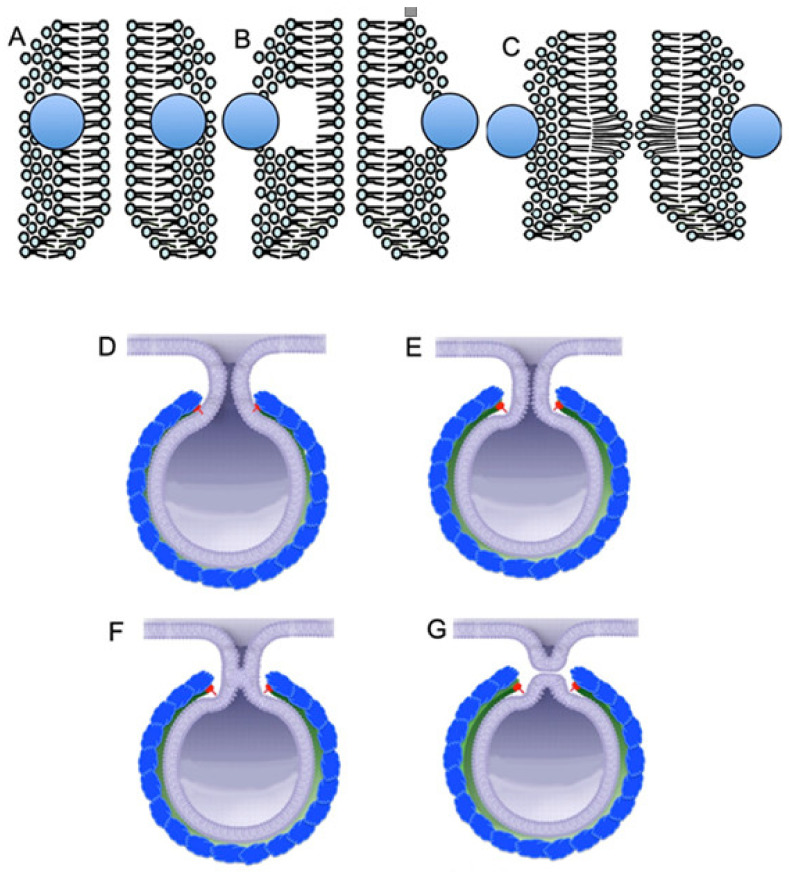
The sudden elimination of coat proteins from the cytosolic leaflet of the lipid membrane can induce destabilization of a thin tubule, due to formation of protrusion of the luminal leaflet (**A**,**B**). Such a quick event induces formation of the excess of the luminal leaflet, and generation of the circular fold on the inner surface of the tube. This fold induces periplasmic fusion and fission of the tube at this site (**C**). (**D**–**G**) Possible model of vesicle fission. (**D**–**G**) Consecutive stages of the fission process. The coat is attached to the neck of the bud (**D**). The thinning of the bud neck induces detachment of red protein from the cytosolic leaflet (**E**). This induces periplasmic fusion and fission of the tube at this site (**F**,**G**).

**Figure 4 ijms-21-07594-f004:**
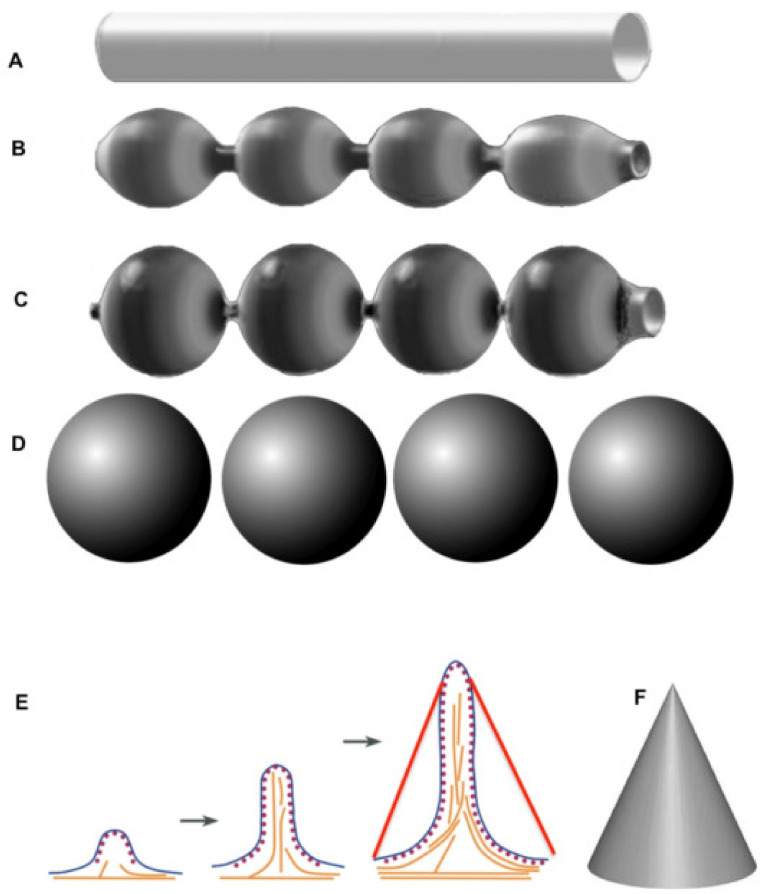
Model describing the mechanism of transformation of a smooth tubule into a row of vesicles. (**A**–**D**) Consecutive stages of transformation of a cylinder into several spheres, with the preservation of surface area and TAA. When the surface area and TAA of a tubule is equal to the sum of the surface areas of the vesicles the radius of vesicle should be 2.2-fold higher than the radius of the cylinder (tubule). (**E**) Consecutive stages of the filopodia growth. The surface area of the cone (**F**) formed if the membrane was not attached to the actin bundle with ezrin, the surface area of the cone (red lines) would be larger than the surface area of the filopodia membrane and the ring of the plasmalemma around filopodia.

**Figure 5 ijms-21-07594-f005:**
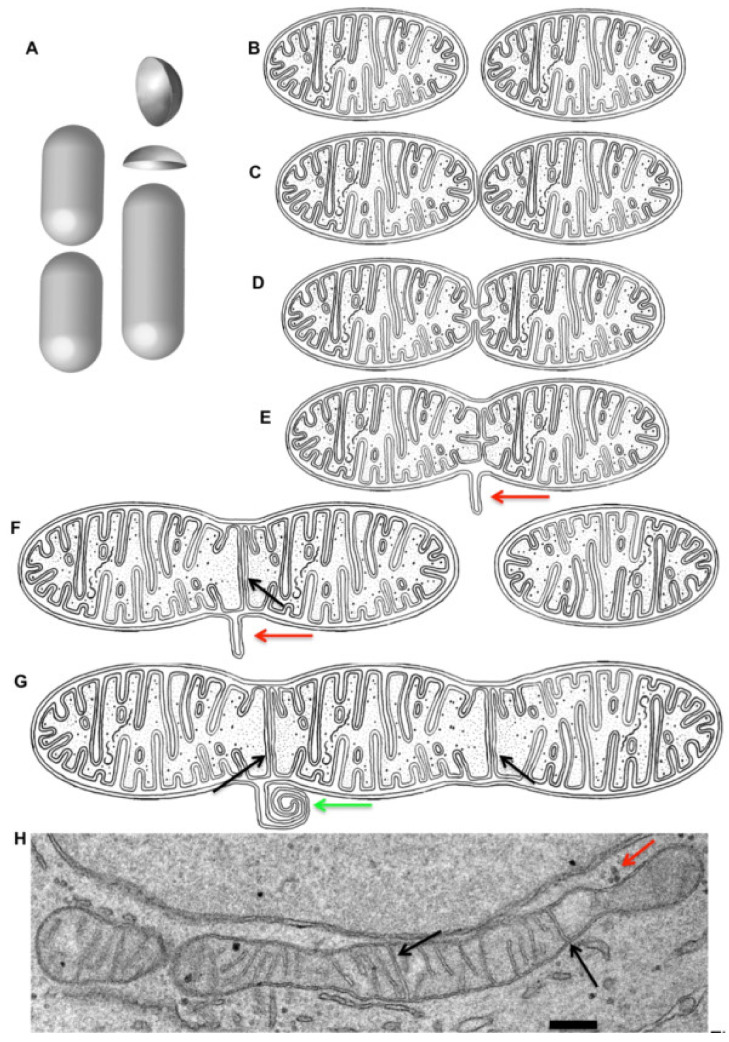
After mitochondria fusion the outer mitochondrial membrane becomes excessive. (**A**) Surface area of two ovoid figures before and after their fusion. Two half-spheres are excessive. (**B**–**G**) Stages of two consecutive mitochondria fusions. The fusion leads to the formation of excess of the surface area of the outer mitochondrial membrane, with the formation of the membrane protrusion (red arrow), which is then transformed into the multi-lamellar organelle (green arrow). Immediately after mitochondrial fusion the wall between the two mitochondria matrix compartments, derived from two mitochondria, is preserved (black arrows in **F**,**G**). (**H**) Example of long mitochondrion with such walls (black arrows). Thus, a mitochondrion formed after several consecutive fusions contains one or several partition walls (black arrows). Bar: 280 nm (**H**).

**Figure 6 ijms-21-07594-f006:**
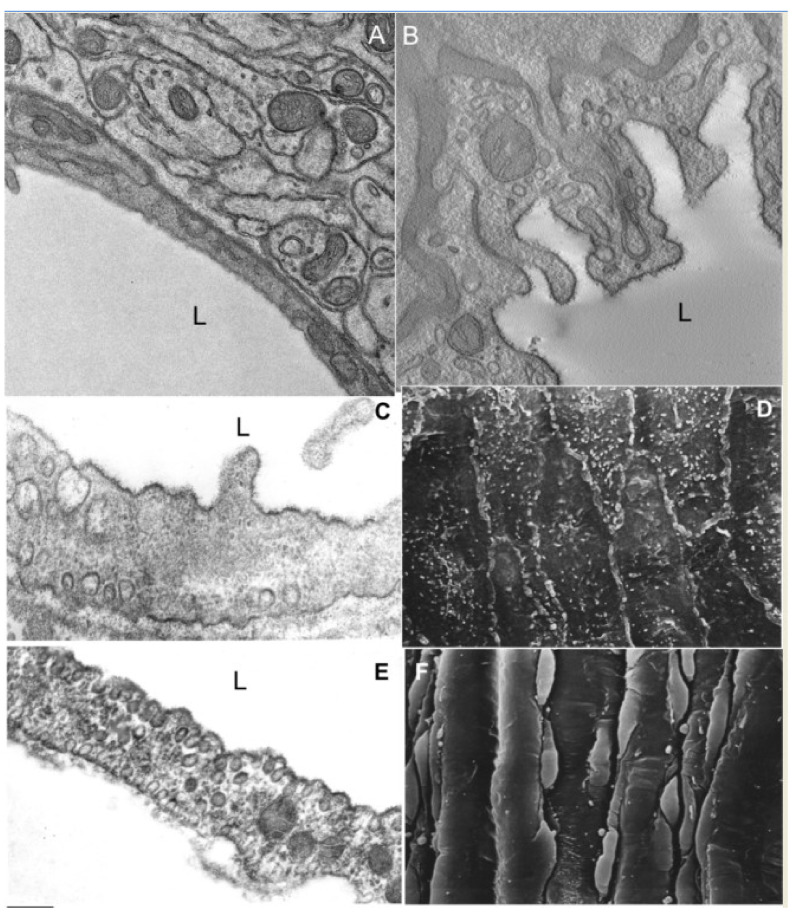
Caveolae and membrane geometry (**A**). (**B**) Low number of caveolae in an endothelial cell (EC) of the brain blood capillary. (**A**) Routine EM section. (**B**) Tomography virtual section. There are almost no caveolae. (**C**) Routine EM section of endothelial a cell of rat aorta after perfusion fixation. (**D**) Scanning EM (SEM) of the surface of endothelial cells (ECs) of rat aorta after perfusion fixation. The number of caveolae is low. (**E**) Routine longitudinal EM section of the aorta EC after immersion fixation. Many caveolae and their derivates are visible. (**F**) SEM of the rat aorta after immersion fixation. L-lumen of capillaries. Bars: 290 nm (**A**); 280 nm (**B**,**C**,**E**); 3 µm (**D**,**F**).

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
