# Peer review of "Membrane Curvature, Trans-Membrane Area Asymmetry, Budding, Fission and Organelle Geometry"

_ijms, 2020, doi:10.3390/ijms21207594_

Round 1

Reviewer 1 Report

The review is of interest and authors improved their manuscript . However, the reading of the revised version of this manuscript to graps the overall concept could be greatly facilitated by a better reorganization of the introduction part, for instance, and by simplifying sentences that are rather long and complex. Authors may want to reinforce the link between their different paragraphs.

I also would like to highlight that the concept of protein concentration as a way to modify the membrane curvature especially in the frame of the Golgi, the main sorting station platform in the cell, is really interesting and should have been more discussed. 

Author Response

Reviewer 1

(x) English language and style are fine/minor spell check required

#We additionally checked our English

Comments and Suggestions for Authors

1. Authors have taken into account all my suggestions. They have to rephrase the sentence at row 50-53; there some mistake and it is not clear.

#We changed this part of the text. Also we made slightly shorter the last part of the text.

Reviewer 2 Report

Authors have taken into account all my suggestions. They have to rephrase the sentence at row 50-53; there some mistake and it is not clear.

Author Response

(x) Moderate English changes required

#We additionally checked our English.

Comments and Suggestions for Authors

  1. The review is of interest and authors improved their manuscript. However, the reading of the revised version of this manuscript to grapes the overall concept could be greatly facilitated by a better reorganization of the introduction part, for instance, and by simplifying sentences that are rather long and complex.

#We made sentences simpler.

  1. Authors may want to reinforce the link between their different paragraphs.

#We corrected these links.

  1. I also would like to highlight that the concept of protein concentration as a way to modify the membrane curvature especially in the frame of the Golgi, the main sorting station platform in the cell, is really interesting and should have been more discussed. 

#We highlighted this part and introduce some new sentences about this.

This manuscript is a resubmission of an earlier submission. The following is a list of the peer review reports and author responses from that submission.

Round 1

Reviewer 1 Report

This is a nice article reviewing the current knowledge the role of trans membrane area asymmetry (TAA) in modulating the geometrical properties of membranes and organelles, discussing different process in which the TAA might be critical (formation of Golgi carriers, MVBs, filopodia etc).

This review is interesting and could be very informative for the readers of International Journal of Molecular Sciences.

Major comments

1) Even though the review describes main progress of the field, authors should balance opinion from other scientists as well as citations. I found that the authors often cite their own review articles. Recent findings showing the contribution of protein in determining the membrane curvature and membrane fission should be debated (for instance, Copic et al 2012. “ER cargo properties specify a requirement for COPII coat rigidity mediated by Sec13p”.Science; Derganc et al 2013. “Membrane bending: the power of protein imbalance.” Trends Biochem Sci.; Zhukovsky et al, 2019 “Protein Amphipathic Helix Insertion: A Mechanism to Induce Membrane Fission” Front in Cell and Dev Biology).

2) Authors critically discussed, in mechanistic terms, the physical properties of Golgi complex and Golgi carriers; however I feel that the biological relevance is missed. Which is the role of curvature and phase transition in lipid sorting and fission of membranes? How is relevant for protein sorting since Golgi complex is a central sorting station? These aspects should be better discussed.

3) Citations should be upgraded. For instance, there are more recent reviews regarding the role of phosphoinositides and other lipids in the Golgi structure and function (e.g., Mayinger 2009, Semin Cell Dev Biology; D’angelo 2012, Sub Biochem; De Matteis et al, 2013 Bioessays; Yang et al, 2008 Nat Cell Biol; Zhukovsky et al, 2019 Front in Cell and Dev Biology).

4) The manuscript should be improved in terms of writing and presentation of the concepts. I found fragmented the way to present the data, and often the same concept is repeated several times in sparse order, therefore loosing the main message (e.g.; the role of proteins is discussed more times, but missing the final conclusive message; I have similar feeling for paragraph 2 where authors firstly discussed on ER-Golgi carriers, then the role of TAA in Golgi structure without linking critically all the items).

Finally, I found hard the reading of the manuscript because of a lot of mistakes and odd terms (see minor comments)

Minor comments:

1) There are many misspellings and typographical errors (some are listed below), a very careful revision is required.

- In the abstract: “exptremely”

- In paragraph 1:

“biologcy” (first row);

“bilayer” (4th row);

“membrane” (17th row; here it should be corrected the misspelling and change in “…of most of cellular membranes..”)

“bilayerbut” (page 2)

“they should be are maximally bent” (page 2)

As compensations,t here could be enlqrgement (page 3)

- In paragraph 2: “complicated”; “COPIcoated” (Pag 4)

- Figure legend 2 have a lot of mistakes

….and so on

2) Spell the abbreviation GC the first time (pag 3)

3) What is GA? Maybe is GC.

4) What does mean “smerovanie”

5) Reference 73 refers a study performed in MDCK cells, not in enterocytes.

6) For more clarity, panel G and H of Figure 1 should be moved in a separate figure that has to appear in according to the order of text.

Author Response

  1. Reply

    In order to illustrate better our hypothesis we replaced some images with new ones. We have re-written more than half of the text and changed the title. Therefore it was useless to label our changes. Also we would like to stress that this text is just another way of explanation of already published results.

    Reviewer 1

    This is a nice article reviewing the current knowledge the role of trans membrane area asymmetry (TAA) in modulating the geometrical properties of membranes and organelles, discussing different process in which the TAA might be critical (formation of Golgi carriers, MVBs, filopodia etc.). This review is interesting and could be very informative for the readers of International Journal of Molecular Sciences.

    # Thanks a lot.

    Major comments

    1) Even though the review describes main progress of the field, authors should balance opinion from other scientists as well as citations. I found that the authors often cite their own review articles. Recent findings showing the contribution of protein in determining the membrane curvature and membrane fission should be debated (for instance, Copic et al 2012. “ER cargo properties specify a requirement for COPII coat rigidity mediated by Sec13p”. Science; Derganc et al 2013. “Membrane bending: the power of protein imbalance.” Trends Biochem Sci.; Zhukovsky et al, 2019 “Protein Amphipathic Helix Insertion: A Mechanism to Induce Membrane Fission” Front in Cell and Dev Biology).

    # We included these references in the paper and added additional new information and eliminated quoting our reviews being focused now on our original papers. However, Paper by Copic et al., describing molecular mechanisms of COPII does not have any relationship to the focus of our review. Zhukovsky and Derganc are included into the reference list.

    2) Authors critically discussed, in mechanistic terms, the physical properties of Golgi complex and Golgi carriers; however I feel that the biological relevance is missed.

    # We added some sentences about the biological relevance

    Which is the role of curvature and phase transition in lipid sorting and fission of membranes? How is relevant for protein sorting since Golgi complex is a central sorting station? These aspects should be better discussed.

    # We re-wrote these parts of the text.

    3) Citations should be upgraded. For instance, there are more recent reviews regarding the role of phosphoinositides and other lipids in the Golgi structure and function (e.g., Mayinger 2009, Semin Cell Dev Biology; D’angelo 2012, Sub Biochem; De Matteis et al, 2013 Bioessays; Yang et al, 2008 Nat Cell Biol; Zhukovsky et al, 2019 Front in Cell and Dev Biology).

    #We added these papers into the reference list, although these papers have no significant importance for our topic. Moreover most of these papers were examined in the reviews by Kozlov.

    4) The manuscript should be improved in terms of writing and presentation of the concepts. I found fragmented the way to present the data, and often the same concept is repeated several times in sparse order, therefore loosing the main message (e.g.; the role of proteins is discussed more times, but missing the final conclusive message; I have similar feeling for paragraph 2 where authors firstly discussed on ER-Golgi carriers, then the role of TAA in Golgi structure without linking critically all the items).

    # We changed the text in order to fulfil the demands by the reviewer.

    Finally, I found hard the reading of the manuscript because of a lot of mistakes and odd terms (see minor comments)

    # We corrected mistakes.

    Minor comments:

    1) There are many misspellings and typographical errors (some are listed below), a very careful revision is required.

    - In the abstract: “exptremely”

    - In paragraph 1:

    “biologcy” (first row);

    “bilayer” (4th row);

    “membrane” (17th row; here it should be corrected the misspelling and change in “…of most of cellular membranes..”)

    “bilayerbut” (page 2)

    “they should be are maximally bent” (page 2)

    As compensations,t here could be enlqrgement (page 3)

    - In paragraph 2: “complicated”; “COPIcoated” (Page 4)

    - Figure legend 2 have a lot of mistakes

    ….and so on

    # We corrected these mistakes

    2) Spell the abbreviation GC the first time (page 3)

    # We added this abbreviation.

    3) What is GA? Maybe is GC.

    # We corrected this abbreviation (it should be GC).

    4) What does mean “smerovanie”

    # We corrected this.

    5) Reference 73 refers a study performed in MDCK cells, not in enterocytes.

    # We corrected this mistake.

    6) For more clarity, panel G and H of Figure 1 should be moved in a separate figure that has to appear in according to the order of text.

    # We agree and corrected images according to the proposal by the reviewer.

    Reviewer 2

Reviewer 2 Report

In this review article, authors intend to explain shape transitions in various organelles from a physics point of view, mainly pivoting on their previously proposed “TAA”-related hypothesis. I agree that geometrical changes of biological membranes cannot exist beyond basic principle of physics (i.e. physical properties of lipids and proteins). Authors tended to use ideal shapes and scenarios to regard biological membranes (e.g. “when organelle volume and surface are constant p.3”), and thus failed to provide a balanced and comprehensive view. To me, TAA is just a physical parameter that is associated with membrane geometry, not necessary a driving/causal factor. To achieve topological change, cells/organelles have to accommodate surface and volume with real membrane materials (by asymmetrical insertion or removal), while TAA is just a ratio. Thus, some hypothetical ideas that authors brought up are imaginary. All in all, I am not convinced that it is suitable for publication as a review. I herein provide a few more detailed reasons:

First of all, this is a reader-unfriendly manuscript: full of typos, grammar errors and long/unstructured sentences without proper punctuations; in this regard, I find it very difficult to understand authors correctly.

Authors intend to include most membranous organelles to discuss how “TAA regularities” could affect their shaping/remodeling behaviors, but in a poorly-structured manner, thus overall the manuscript sounds busy and messy. Some ideas are repeatedly mentioned in different sections. Author should have clearly stated/defined the relation and difference between TAA and membrane curvature, not just simply mingling these two concepts. Membrane curvature can describe the shape at any point of a surface, while TAA appears to consider the structure/compartment as a whole (i.e. vesicle, tube or disk etc.,  based on calculation examples that authors provided)?

Besides, ideas or context often jump abruptly without clear logical relevance or enough background information/explanations. For instance, authors just pointed out a role of “phospholipid modifying enzymes in maintaining Golgi structure (on p.6)”. In the next sentence, they highlighted the idea that “golgi networks have a lower tension than ER”. Some unnecessary details confuse readers from understanding authors’ major points: e.g. sample preparation for EM increases TAA (on page 1, what is the point here?) etc. In addition, most figures (e.g. Fig.1) lack sufficient commentary in either context or legends.

My major problem with this manuscript is that key claims are mostly speculative and appear baseless. Many statements are scientifically incorrect or lacking necessary evidence. I just copy a small portion of such examples as follows:

“TAA is proportional to the total membrane curvature, it is the largest in vesicles and the smallest in flat membranes and saddle-like membrane in the rims of the perforations. (p.1)” –As defined by authors, TAA of flat membrane is 1 and TAA of saddle-like membrane is smaller than 1. So, “largest” or “smallest” is improper used here; also TAA of vesicles differs as diameters vary.

“If the thickness of the membrane is 10 nm, the difference between the areas of the outer layer and the inner layer will be higher. (p.1)” – for a flat membrane, TAA will remain the same regardless of 4nm or 10nm thickness, right?

“The crowded protein environment on the surface of cellular membranes can contribute to membrane shape change [20]. However, it seems that flippases and even simple flip-flops are more important [21, 22] (p.2)” – more important for what? For contributing to membrane shape change or for cell physiology?

“COPI/BARS machine can induce fission. Vesicles can't turn into a disk. They can turn into a cylinder or torroid with a smaller diameter than vesicles. p.3” –no reference and evidence for these statements.

“This excess of the outer or external membrane of mitochondria could be used for the formation of multilayered organelle (MLO; Fig. 1G). MLOs are constantly formed during fusion of mitochondria and then fission. When mitochondria undergo fission it is necessary to deliver similar amount of membrane to the site of fission. This operation is performed by the tubule of the ER, which surrounds mitochondria near the site of fission [80]. These contacts between mitochondria and ER could provide lipids necessary for the fission. p.15” –Whether or not lipid transport occurs at ER-mito contacts is still debatable in the field, let alone its necessity in mitochondrial fission. This claim is too speculative. Also, I cannot find evidence supporting the claim “MLOs are constantly formed during fusion of mitochondria”.

Author Response

Reviewer 2

English language and style

(x) Extensive editing of English language and style required

#We asked the Englishman to correct our English.

In this review article, authors intend to explain shape transitions in various organelles from a physics point of view, mainly pivoting on their previously proposed “TAA”-related hypothesis. I agree that geometrical changes of biological membranes cannot exist beyond basic principle of physics (i.e. physical properties of lipids and proteins).

#Dear Reviewer, than you very much

Authors tended to use ideal shapes and scenarios to regard biological membranes (e.g. “when organelle volume and surface are constant p.3”), and thus failed to provide a balanced and comprehensive view.

# In general we agree. There could be a lot of shape variations and we illustrated this by our fist Figure showing the undulation of cisterna rims could increase TAA without changes in the volume and even in the surface area. However, data obtained after high pressure freezing indicate that in most cases organelles are in the geometrical shape close to the ideal one (we added the images from our studies, which demonstrate this. We also added into the text some words describing this limitation.

To me, TAA is just a physical parameter that is associated with membrane geometry, not necessary a driving/causal factor. To achieve topological change, cells/organelles have to accommodate surface and volume with real membrane materials (by asymmetrical insertion or removal), while TAA is just a ratio.

# We changed the concept of TAA and replaced it by curvature.

Thus, some hypothetical ideas that authors brought up are imaginary. All in all, I am not convinced that it is suitable for publication as a review.

# We changed the classification of the paper and indicated that this is a hypothesis.

I herein provide a few more detailed reasons:

 First of all, this is a reader-unfriendly manuscript: full of typos, grammar errors and long/unstructured sentences without proper punctuations; in this regard, I find it very difficult to understand authors correctly.

# We asked Englishman to check the text and corrected these mistakes.

Authors intend to include most membranous organelles to discuss how “TAA regularities” could affect their shaping/remodelling behaviors, but in a poorly-structured manner, thus overall the manuscript sounds busy and messy. Some ideas are repeatedly mentioned in different sections. Author should have clearly stated/defined the relation and difference between TAA and membrane curvature, not just simply mingling these two concepts. Membrane curvature can describe the shape at any point of a surface, while TAA appears to consider the structure/compartment as a whole (i.e. vesicle, tube or disk etc., based on calculation examples that authors provided)?

# We corrected this.

Besides, ideas or context often jump abruptly without clear logical relevance or enough background information/explanations. For instance, authors just pointed out a role of “phospholipid modifying enzymes in maintaining Golgi structure (on p.6)”. In the next sentence, they highlighted the idea that “golgi networks have a lower tension than ER”.

# This conclusion is taken for the paper by Jennifer Lipoincott-Schwartz. In any case we corrected these details.

Some unnecessary details confuse readers from understanding authors’ major points: e.g. sample preparation for EM increases TAA (on page 1, what is the point here?) etc. In addition, most figures (e.g. Fig.1) lack sufficient commentary in either context or legends.

# We explained why preparation details are important. The reason is that when the scientist measures curvature the results would depend on the fixation and staining procedure. For example, the thickness of biological membrane is only 4.3 nm, whereas the thickness of membrane image after treatment with OsO4 is 9 nm.

My major problem with this manuscript is that key claims are mostly speculative and appear baseless.

# We explained our hypotheses better.

Many statements are scientifically incorrect or lacking necessary evidence. I just copy a small portion of such examples as follows:

 “TAA is proportional to the total membrane curvature, it is the largest in vesicles and the smallest in flat membranes and saddle-like membrane in the rims of the perforations. (p.1)” –As defined by authors, TAA of flat membrane is 1 and TAA of saddle-like membrane is smaller than 1. So, “largest” or “smallest” is improper used here; also TAA of vesicles differs as diameters vary.

#We explained this better and corrected such statements.

“If the thickness of the membrane is 10 nm, the difference between the areas of the outer layer and the inner layer will be higher. (p.1)” – for a flat membrane, TAA will remain the same regardless of 4nm or 10nm thickness, right?

# We added the explanation why the thickness of membrane in EM samples is 9–10 nm whereas in reality its thickness is only 4.3 nm and even smaller. The reason is the measurement of the curvature. We also added a figure explaining this.

“The crowded protein environment on the surface of cellular membranes can contribute to membrane shape change [20]. However, it seems that flippases and even simple flip-flops are more important [21, 22] (p.2)” – more important for what? For contributing to membrane shape change or for cell physiology?

# We explained these points better.

“COPI/BARS machine can induce fission. Vesicles can't turn into a disk. They can turn into a cylinder or toroid with a smaller diameter than vesicles. p.3” –no reference and evidence for these statements.

# We rewrote this part.

“This excess of the outer or external membrane of mitochondria could be used for the formation of multilayered organelle (MLO; Fig. 1G). MLOs are constantly formed during fusion of mitochondria and then fission. When mitochondria undergo fission it is necessary to deliver similar amount of membrane to the site of fission. This operation is performed by the tubule of the ER, which surrounds mitochondria near the site of fission [80]. These contacts between mitochondria and ER could provide lipids necessary for the fission. p.15” –Whether or not lipid transport occurs at ER-mito contacts is still debatable in the field, let alone its necessity in mitochondrial fission. This claim is too speculative.

#We added detailed explanation of this consequence. Indeed, during cells life mitochondria constantly fuse and then undergo fission. It is well-established process (we added the reference) and we added papers where this is well described. Due to constant fusion the excess of OMM is generated and under normal conditions mitochondria derived vesicles MDVs are formed from this additional membrane of OMM and these vesicles (in fact MLOs) go to lysosomes. Transport of fatty acids through ER-Mito contacts is shown by Rambold et al. (2011, 2015) after mitochondria fusion when excess of OMM is formed. Now it is almost a consensus that free fatty acids cannot travel through cytosol if the membranes are separated (Rambold et al., 2015; doi: 10.1016/j.devcel.2015.01.029; Stefan et al., 2017; doi: 10.1186/s12915-017-0432-0; Cohen et al., 2018. doi: 10.1080/23723556.2015.1043038). For instance, there is no background of RedC12 in Fig. 1B by Rambold et al. (2015). Moreover, pulling one cholesterol molecule (or free fatty acid) out of a lipid bilayer into water required an energy barrier of 80–90 kJ/mol corresponding to hydrolysis of about 1.5 ATP molecules. Thus, only lipid transfer proteins can deliver cholesterol and free fatty acids from membrane of lysosomes into membrane of the ER. For this it is also necessary to have transfer of cholesterol and FFAs from the luminal leaflet of lysosomal membrane to its cytosolic monolayer. Numerous (now more than 131) lipid- and sterol-transfer proteins have been identified to mediate directional cholesterol transfer at membrane contact sites formed between two closely apposed organelles (reviewed by Mironov et al., 2020). In any case we eliminated this statement.

Also, I cannot find evidence supporting the claim “MLOs are constantly formed during fusion of mitochondria”.

# It is shown here (Mironov et al., 2017. DOI: 10.1080/01913123.2016.1269416. Also it is submitted to Cell).

Round 2

Reviewer 2 Report

The language and structure have been greatly improved. But there are still a handful of typos, grammar errors, long/unstructured sentences and citation errors that need further correction. Having said that, I am still not convinced that it is suitable for publication as a review. Authors did not provide the clear definition of “TAA membrane curvature hypothesis” and how this hypothesis is related to predictions that mentioned in the manuscript. Again, to me, TAA is a physical parameter that is associated with membrane geometry, not necessary a causal factor. But it can be potentially an interesting viewpoint. Authors may consider it an Opinion article and further tune down their claims.

Asymmetrical insertion or removal of lipids and proteins can contribution to changes of TAA in a quantitative fashion. But authors tend to interpret cell biology observations using ‘TAA hypothesis’ in a rather qualitative manner. How many proteins or lipids should detach/insert to achieve enough change of TAA? How significant should TAA change to trigger “compensative outcomes”, as fission/fusion events? Without these borderlines, one can hardly accept that TAA can be a driving factor for shape transition. Regarding organelle shapes, vesicles are close to sphere, but morphologies of the ER and mitochondria are far more complicated (not as simple as cylinders; e.g. their luminal diameters vary).

My Response to following comments

#We added detailed explanation of this consequence. Indeed, during cells life mitochondria constantly fuse and then undergo fission. It is well-established process (we added the reference) and we added papers where this is well described. Due to constant fusion the excess of OMM is generated and under normal conditions mitochondria derived vesicles MDVs are formed from this additional membrane of OMM and these vesicles (in fact MLOs) go to lysosomes. Transport of fatty acids through ER-Mito contacts is shown by Rambold et al. (2011, 2015) after mitochondria fusion when excess of OMM is formed. Now it is almost a consensus that free fatty acids cannot travel through cytosol if the membranes are separated (Rambold et al., 2015; doi: 10.1016/j.devcel.2015.01.029; Stefan et al., 2017; doi: 10.1186/s12915-017-0432-0; Cohen et al., 2018. doi: 10.1080/23723556.2015.1043038). For instance, there is no background of RedC12 in Fig. 1B by Rambold et al. (2015). Moreover, pulling one cholesterol molecule (or free fatty acid) out of a lipid bilayer into water required an energy barrier of 80–90 kJ/mol corresponding to hydrolysis of about 1.5 ATP molecules. Thus, only lipid transfer proteins can deliver cholesterol and free fatty acids from membrane of lysosomes into membrane of the ER. For this it is also necessary to have transfer of cholesterol and FFAs from the luminal leaflet of lysosomal membrane to its cytosolic monolayer. Numerous (now more than 131) lipid- and sterol-transfer proteins have been identified to mediate directional cholesterol transfer at membrane contact sites formed between two closely apposed organelles (reviewed by Mironov et al., 2020). In any case we eliminated this statement.

Here is one such example of my major concerns with this manuscript. Observations of MDVs or MLOs are real. But, none of these studies that authors cited (including Mironov et al., 2017. DOI: 10.1080/01913123.2016) showed that formation of MDVs or MLOs is due to constant fusion of mitochondria. It can be totally irrelevant. Authors can speculate it, but they cannot overinterpret in favour of own ideas by mingling facts with speculations.

The same problem goes for mitochondrial fission. There are lipid transferring proteins at membrane contact sites (e.g. ER-mito, mito-lyso). Whether or not these proteins locally function to transport lipids is unclear (reviewed in doi:10.1126/science.aan5835 refer to Section #Factors and Functions of ER-Mitochondria MCSs). Even they do, it does not mean that these lipids are required for mitochondrial fission. Even they are needed for fission, it does not mean it is to compensate TAA change.